# Coronary Microcirculation: The Next Frontier in the Management of STEMI

**DOI:** 10.3390/jcm12041602

**Published:** 2023-02-17

**Authors:** Dejan Milasinovic, Olga Nedeljkovic, Ruzica Maksimovic, Dragana Sobic-Saranovic, Djordje Dukic, Vladimir Zobenica, Dario Jelic, Milorad Zivkovic, Vladimir Dedovic, Sanja Stankovic, Milika Asanin, Vladan Vukcevic

**Affiliations:** 1Department of Cardiology, University Clinical Center of Serbia, 26 Visegradska, 11000 Belgrade, Serbia; 2Faculty of Medicine, University of Belgrade, 11000 Belgrade, Serbia; 3Center for Radiology and Magnetic Resonance, University Clinical Center of Serbia, 11000 Belgrade, Serbia; 4Center for Nuclear Medicine with PET, University Clinical Center of Serbia, 11000 Belgrade, Serbia; 5Center for Medical Biochemistry, University Clinical Center of Serbia, 11000 Belgrade, Serbia; 6Faculty of Medical Sciences, University of Kragujevac, 34000 Kragujevac, Serbia

**Keywords:** acute myocardial infarction, percutaneous coronary intervention, coronary microvascular dysfunction, coronary flow reserve

## Abstract

Although the widespread adoption of timely invasive reperfusion strategies over the last two decades has significantly improved the prognosis of patients with ST-segment elevation myocardial infarction (STEMI), up to half of patients after angiographically successful primary percutaneous coronary intervention (PCI) still have signs of inadequate reperfusion at the level of coronary microcirculation. This phenomenon, termed coronary microvascular dysfunction (CMD), has been associated with impaired prognosis. The aim of the present review is to describe the collected evidence on the occurrence of CMD following primary PCI, means of assessment and its association with the infarct size and clinical outcomes. Therefore, the practical role of invasive assessment of CMD in the catheterization laboratory, at the end of primary PCI, is emphasized, with an overview of available technologies including thermodilution- and Doppler-based methods, as well as recently developing functional coronary angiography. In this regard, we review the conceptual background and the prognostic value of coronary flow reserve (CFR), index of microcirculatory resistance (IMR), hyperemic microvascular resistance (HMR), pressure at zero flow (PzF) and angiography-derived IMR. Finally, the so-far investigated therapeutic strategies targeting coronary microcirculation after STEMI are revisited.

## 1. Introduction

Widespread adoption of timely invasive reperfusion strategies over the last two decades has significantly improved the prognosis of patients with ST-elevation myocardial infarction (STEMI), making primary percutaneous coronary intervention (PCI) the standard of care [1]. Although time delay between symptoms onset and primary PCI has initially been recognized as a key variable leading to mortality reduction, real-world registry data demonstrated a lack of mortality benefit just by increasing the proportion of STEMI patients meeting the guideline-recommended thresholds for timely reperfusion [2]. Therefore, it is today important to understand the sources of residual risk after primary PCI, which may be divided into the following domains: (a) comorbidities that limit life expectancy, (b) the extent of coronary artery disease (CAD) in both culprit and non-culprit territories, together with the associated risk of revascularization procedures, and (c) myocardial remodeling leading to left ventricular (LV) dysfunction, heart failure (HF) and the risk of sudden cardiac death (SCD). Focusing on the last point, several pathophysiological processes have been described to explain the extent of myocardial injury and impaired LV function after STEMI despite timely and successful epicardial flow restoration, including myocardial reperfusion injury [3] and no-reflow phenomenon [4]. Whereas the concept of myocardial reperfusion injury has mainly been understood in terms of different molecular pathways leading to cardiomyocyte death following myocardial reperfusion after a period of ischemia, no-reflow phenomenon denotes the absence of adequate reperfusion at the level of coronary microcirculation, thus leaving areas of the myocardium underperfused despite optimal epicardial flow [4]. Given the experimental data on the extension of microvascular damage and infarction territory following reperfusion [5], protecting the reperfused myocardium has been seen as an opportunity to improve outcomes after STEMI [6]. However, the exact pathophysiological pathways linking reperfusion injury to microvascular damage and cardiomyocyte death remain debated [7]. From the practical perspective of contemporary invasive reperfusion strategies, coronary microcirculation acts as a downstream flow controller, making it imperative to understand the prevalence, causes and clinical consequences of inadequate microvascular reperfusion after successful recanalization of the infarct-related artery (IRA). Thus, our aim is to review the hitherto collected evidence on the occurrence of coronary microvascular dysfunction (CMD) following primary PCI, the means of assessment and its association with the infarct size (IS) and clinical outcomes. Finally, we will revisit therapeutic strategies targeting coronary microcirculation that have so far been tested in patients with STEMI. 

## 2. Frequency and Prognosis of CMD in Patients with STEMI

Although contemporary percutaneous techniques are associated with successful recanalization of the occluded epicardial IRA in 95% of cases, the occurrence of CMD following primary PCI has been consistently reported at ≈50%, from the early days of reperfusion [8] and into the modern era of widespread primary PCI [9]. These observations suggest that the mechanisms leading to microvascular dysfunction have largely remained unaffected by improvements in epicardium-centered PCI techniques over time. Several mechanisms of inadequate reperfusion at the level of coronary microcirculation have been described (Figure 1), including a general process of microvascular injury (MVI) involving the swelling and rupture of endothelial cells, but also more specifically, microvascular obstruction (MVO) and intramyocardial hemorrhage (IMH) [7]. MVO was described on cardiac magnetic resonance (CMR) and it referred to contrast-devoid areas within the contrast-enhanced infarction zone, thus suggesting the obstruction of coronary microcirculation due to downstream microvascular embolization, plugging and extravascular compression. Experimental studies pairing histopathology with CMR imaging showed that the contrast-devoid area within the infarction zone may also signify the destruction of coronary microvasculature, i.e., MVI, with exiting of erythrocytes and consequent intramyocardial hemorrhage [10]. Unlike MVO, which can regress over time after STEMI, intramyocardial hemorrhage represents an irreversible consequence of microvascular destruction following reperfusion [7]. Importantly, MVO on CMR has been shown to independently predict mortality and rehospitalization for heart failure during the first year after STEMI [11]. A more practical approach to diagnosing CMD in the setting of primary PCI is the invasive assessment of coronary flow reserve (CFR) and microvascular resistance following the reestablishment of the epicardial flow. Invasive approaches have the advantage of timely CMD diagnosis leading to risk stratification at the end of primary PCI and allowing for early commencement of potential therapeutic strategies that may target CMD. Importantly, microvascular damage following primary PCI has been associated with impaired short and long-term clinical outcomes regardless of the diagnostic method [12,13,14,15]. Although the exact mechanisms are not clear, one possible pathway is by impacting infarct size, which has been shown to have a strong association with the development of chronic heart failure and ultimately mortality following STEMI [16]. That being said, it should be noted, not only that microvascular injury predicts infarct size [17,18], but that its presence after STEMI is also associated with an increased risk of death, reinfarction and heart failure above and beyond the infarct size, making it an independent therapeutic target [9,14]. 

Taking note of the clinical utility of this accumulated knowledge about the frequency and the prognostic relevance of CMD after STEMI, we further address (1) the potential of invasive indices of microvascular function to risk stratify patients immediately after successful primary PCI in the catheterization laboratory, and (2) the results of the so far tested therapies targeting CMD in patients with STEMI. 

## 3. Invasive Assessment of CMD following Primary PCI

Although TIMI flow <3 is intuitively associated with impaired reperfusion at the level of microcirculation and it is a recognized predictor of adverse events following primary PCI [19], the very low rate of such angiographically documented no-reflow of <5% is not commensurate with the findings from both experimental and clinical studies with the evidence of CMD in half of patients with STEMI having TIMI 3 at the end of the procedure [9,20]. Hence, using TIMI classification to describe epicardial flow and by extension estimate the presence of CMD may be the accessible tool in the catheterization laboratory, albeit grossly underestimating its true frequency. Myocardial blush grade (MBG) that relies on the assessment of angiographic contrast density in the distal territory of an epicardial artery has been used as a surrogate of the microvascular reperfusion, and it has been shown to predict mortality after primary PCI independently of the TIMI flow [21]. Although MBG did correlate to a certain degree with CMR-based MVO, much like the other angiographic surrogate, TIMI flow, it still failed to detect a considerable portion of patients with signs of microvascular injury on CMR [20,22]. Going beyond angiographic markers, such as TIMI flow and MBG, invasive assessments of coronary microcirculation have been used to assess microcirculatory function. Recently, increased coronary wedge pressure (CWP > 38 mmHg), as measured through a microcatheter advanced distally to the occlusion site and before culprit artery recanalization, was associated with adverse LV remodeling over 60 months follow-up after STEMI [23]. Wire-based assessments of flow velocity with either a Doppler or thermodilution method after epicardial flow restoration have been shown to correlate well with the CMR-based detection of microvascular damage. In principle, wire-based methods have the capability to measure pressure in the distal coronary artery and flow, either as flow velocity (Doppler) or as mean transit time (thermodilution method), under resting and hyperemic conditions. Combining simultaneous pressure and flow measurements, several indices of microvascular function have been developed (Figure 2) including coronary flow reserve (CFR), index of microcirculatory resistance (IMR), hyperemic microvascular resistance (HMR), instantaneous diastolic flow velocity-pressure slope (IHDVPS) and pressure at zero flow (PzF). 

### 3.1. Coronary Flow Reserve

The concept of CFR is rooted in the observation that coronary flow can be increased under stress conditions. In the catheterization laboratory, CFR is defined as the ratio between Doppler wire-derived average peak flow velocity (APV) or thermodilution-based mean transit time (Tmn) in hyperemia over resting conditions. APV is directly measured by an intracoronary wire equipped with a Doppler crystal at the tip [24], whereas Tmn is calculated based on temperature changes detected by proximal and distal temperature sensors on a dedicated intracoronary wire [25]. Since flow velocity is dependent on both the presence of epicardial disease as well as on microvascular function, CFR values reflect the combined functional status of the epicardial artery and the microcirculation. Paired measurements in the same patient cohort showed moderate agreement between Doppler- and thermodilution-derived CFR, documenting correlation coefficients (r) of 0.60–0.80 and higher variability of thermodilution-derived CFR values (0.9–7.1. vs. 1.1–4.3 for Doppler-derived CFR), albeit with a higher rate of successful data acquisition with the thermodilution-based method (in 84–97% of patients vs. 57–69% with the Doppler-wire) [25,26,27]. When both were head-to-head compared with the gold standard [15O] H2O PET imaging assessment of regional coronary flow, the Doppler-derived method had a closer correlation with PET-derived CFR (r = 0.82), compared with thermodilution (r = 0.55), with a tendency of the latter to overestimate CFR at higher values [27]. Initial studies in patients with STEMI documented that lower CFR values in the reperfused infarct-related artery have been associated with adverse LV remodeling [28,29] and were shown to better correlate with LV functional recovery as compared with angiographic parameters such as TIMI flow and MBG [30]. In terms of clinical events, CFR ≤ 1.3 measured in the infarct-related artery immediately after successful primary PCI was found to be associated with a risk of both acute heart failure and increased rate of heart failure hospitalizations during longer-term follow-up [31]. These early observations were corroborated by long-term clinical follow-up demonstrating a non-significant increase in 10-year mortality in patients with post-primary PCI CFR values < 1.5 in the IRA and a significant increase in mortality in patients with CFR < 2.1 in a reference non-IRA vessel [32]. More recently, CFR < 2 was shown to predict CMR-based MVO with 79% sensitivity, albeit with a low specificity of 34% [15], highlighting the need to understand the predictive capacity of more microcirculation-specific indices such as IMR or HMR. 

### 3.2. Thermodilution-Derived Index of Microcirculatory Resistance 

Whereas CFR is determined by the resistance to flow in both the epicardial and microvascular vessels, IMR and HMR reflect the isolated contribution of microvascular resistance to coronary flow impedance. This is achieved by relating the measured coronary flow velocity to the distal coronary pressure under hyperaemic conditions. IMR is calculated by multiplying the distal coronary pressure with the Tmn during hyperaemia, which is a surrogate of maximum flow. Hyperaemic Tmn is measured as an average of 3 values obtained after repeated boluses of 3 mL of saline injected into the coronary artery through the guiding catheter. Conceptually and unlike CFR, IMR values are not affected by either heart rate, blood pressure, LV contractility, resting flow conditions or the presence of epicardial stenoses if the collateral flow is taken into account [33]. The IMR’s ability to estimate microvascular resistance independently of the degree of epicardial stenosis, when corrected for the collateral flow, was demonstrated in both the experimental [34] and clinical settings [35]. Of note, in the setting of STEMI, IMR is measured after IRA stenting and thus the adjustment for epicardial stenosis is usually not necessary. Increased IMR at the end of primary PCI was found to be related to the occurrence of MVO on CMR [36,37]. A pooled analysis of six studies with overall 288 patients identified that IMR > 41U was associated with a very high likelihood of MVO [38]. Building on these findings, IMR was furthermore shown to correspond to the extent of MVO as expressed by the percent of the left ventricle [39]. Moreover, IMR as a continuous variable and using the dichotomous cut-off value of 40 was shown to predict infarct size following acute MI [40,41]. In line with these findings, IMR was also shown to independently predict the degree of LV remodeling and functional recovery following STEMI [42,43]. Despite the accumulated evidence suggesting a correlation between IMR and MVO, and the association of each of the two with infarct size, a recent study showed discordance between the two in a third of cases and that up to a half of patients with signs of MVO on CMR exhibited IMR values < 40 [44]. Importantly, in patients with both MVO and IMR > 40, final infarct size at 6 months was larger, with an 11-fold increased risk of IS > 25% of LV, a recognized mortality predictor [45], as compared with patients with MVO and IMR < 40. Moreover, infarct size regression over 6 months after primary PCI was more pronounced in patients with IMR < 40, independently of MVO [44]. Overall, IMR was shown to predict infarct size independently of MVO. In a comparative study, IMR was shown to predict MVO, myocardial hemorrhage and infarct size independently of CFR [40]. Moreover, in the same study, IMR > 40 was associated with an increased rate of heart failure hospitalizations and death and CFR was not linked to either clinical outcomes or the above parameters of myocardial injury [40]. Given the described discrepancy in the predictive capacities of CFR and IMR, it seems important to understand their pathophysiological correlates within the broader context of microvascular injury and their respective prognostic value. In this regard, it has been shown that whereas IMR values correlate with the (viable) myocardial mass, CFR is independent thereof [46]. The combined use of CFR and IMR was associated with excellent prognostic ability in terms of predicting MVO on CMR (area under the curve (AUC) 0.941) [36]. All patients with IMR > 36 paired with CFR ≤ 1.7 had MVO on CMR by day 7 after primary PCI, whereas it was not present in any of the patients with IMR ≤ 36 and CFR > 1.7 [36]. 

As IMR is seen to also reflect structural microvascular damage expressed by an impediment to flow at maximum hyperemia, resistive reserve ratio (RRR) assesses the functional ability to increase flow from resting to hyperemic conditions, representing the ratio of basal resistance to IMR. RRR was shown to be lower in STEMI as compared with patients with chronic syndromes [47], reflecting the diminished vasodilatory capacity of microcirculation in STEMI. In terms of its predictive capacity, RRR was associated with the extent of MVO, infarct size and myocardial hemorrhage on CMR, as well as with the occurrence of heart failure hospitalizations after STEMI [40,41]. 

The generation of thermodilution-derived temperature tracings that allow for the calculation of mean transit time provides an opportunity for the off-line analysis of the temperature-tracing patterns and temperature recovery time. Both parameters have been proposed to reflect the status of coronary microcirculation and have been investigated in the setting of STEMI. Thermodilution-derived temperature recovery time (TRT) is measured as the duration between the thermodilution curve nadir at hyperemia and 20% from the body temperature at resting conditions. It was associated with MVO independently of CFR and IMR, and it predicted the occurrence of death and heart failure hospitalization at 5 years after STEMI [48]. Beyond the temperature recovery time, an analysis of the thermodilution curve patterns, including narrow unimodal, wide unimodal and bimodal, seems to provide additional prognostic value. When patients were stratified according to the three described patterns, IMR was increased in those with wide unimodal or bimodal thermodilution curves, whereas MVO was more frequently associated with the bimodal pattern [37]. Over the 5-year follow up, bimodal thermodilution curves predicted the occurrence of death and HF hospitalizations, independently of IMR > 40 [49]. These findings corroborated previous results of several studies documenting the ability of IMR to predict adverse clinical outcomes in patients with STEMI [15,50,51,52]. IMR > 40 was associated with early complications following primary PCI such as cardiogenic shock, pulmonary edema, cardiac rupture, malignant arrhythmias and cardiac death within 30 days, and was better in predicting those when compared with CFR and traditional risk scores such as Zwolle and PAMI-II [50]. Over 3 years, IMR > 40 was associated with a significant increase in rates of death and heart failure hospitalization, independently of CFR and clinical risk factors [51]. Importantly, IMR > 40 predicted the occurrence of death and heart failure above and beyond infarct size [15], making it potentially an independent therapeutic target. 

Taken together, the described data offer solid evidence that thermodilution-derived indices of microvascular function following primary PCI may stratify patients according to the risk of (a) MVO on CMR, (b) larger infarct size and c) death and HF hospitalizations in the follow up (Table 1). 

### 3.3. Doppler-Wire-Derived Hyperemic Microvascular Resistance 

HMR is calculated by dividing the distal coronary pressure by the average peak flow velocity (APV) as measured by the Doppler method at hyperemia. Both pressure and APV are obtained simultaneously from a pressure wire additionally equipped with a flow sensor. HMR was found to modestly correlate with IMR in a mixed population of patients with acute myocardial infarction and stable angina (r = 0.41) [57]. In patients with STEMI, HMR was associated with the occurrence of MVO on CMR, with the suggested cut-off value of 2.5 mmHg/cm/s [58], which corroborated earlier findings demonstrating the correlation of HMR with the transmural extent of MI [17]. In terms of predicting clinical outcomes, HMR > 2.82 was associated with an elevated risk of death and heart failure hospitalizations during the 8-year follow-up [53]. A comparative study indicated a better ability of HMR (with the cut-off value > 3 mmHg/cm/s) over CFR < 1.5 to predict cardiac death or heart failure hospitalization [54]. Additional measures of resistance to flow obtained from the simultaneous pressure and flow recordings by the Doppler wire, which have been investigated as surrogates of microvascular injury in STEMI, including instantaneous hyperemic diastolic flow velocity-pressure slope (IHDVPS) and pressure at zero flow (PzF). These approaches utilize the linear relationship of pressure and flow in the mid-to-late diastole to provide an estimate of microvascular conductance. Although IHDVPS and PzF were shown to correlate with each other in patients with STEMI [58], and both had been previously associated with microcirculatory structural damage, expressed as diminished capillary density and arteriolar obliteration in a myocardial biopsy study [59], PzF only, unlike IHDVPS, has consistently been associated with infarct size and imaging correlates of CMD such as MVO on CMR [17,58,60]. Conceptually, PzF is an extrapolation of the regression line relating pressure to flow in a way that estimates the theoretical pressure value at which intracoronary flow would stop, thus reflecting the intraluminal pressure required to maintain flow against the resistance that takes into account extravascular compression. Given that myocardial edema and hemorrhage are known substrates of CMR-based MVO, PzF represents a measure of CMD that may combine vascular reactivity with accounting for extravascular sources of resistance to flow. Furthermore, PzF was found to strongly correlate with LV filling pressures following acute MI, highlighting the impact of LV function on myocardial reperfusion, which may be explained by elevated intracardial pressure and cardiac stiffness [61]. When compared with CFR, PzF was more closely associated with the degree of myocardial contractility recovery following primary PCI [60]. Importantly, a more recent direct comparison revealed that PzF (AUC = 0.94) had a better capacity to predict large infarct size after STEMI as compared with both HMR (AUC = 0.74) and IMR (AUC = 0.54) [62]. A strength of the study was its primary endpoint being infarct size ≥ 24% of the LV, which is a recognized predictor of mortality following STEMI [45]. PzF cut-off > 42 mmHg was predictive of this primary endpoint [62]. 

### 3.4. Coronary Angiography-Derived Index of Microvascular Resistance

Recent utilization of coronary angiography images to yield an estimation of flow and pressure gradient across the coronary tree provided an accurate surrogate for fractional flow reserve (FFR), without intracoronary instrumentation with pressure/temperature sensor-equipped wires [63]. Further developments in this technology revealed the potential to use coronary angiography to also estimate microvascular resistance [64]. Angiography-derived IMR has so far been investigated in patients with INOCA [65] and acute myocardial infarction [56,66]. A recent study in successfully revascularized STEMI patients (all had TIMI3 post-primary PCI) showed a good correlation between wire-based and angio-derived IMR (r = 0.782, *p* < 0.001) [56]. Moreover, all patients with IMR > 40 also had angio-IMR > 40. In terms of clinical outcomes, angio-IMR > 40 was associated with a significantly higher risk of cardiac death and readmission for heart failure during 10 years of follow-up [56]. 

## 4. Coronary Microcirculation as a Therapeutic Target 

The above-described evidence base testifies to the combined ability of the indices of microvascular dysfunction measured at the end of primary PCI, to predict (a) the occurrence of MVO and myocardial hemorrhage on CMR, (b) infarct size, (c) the degree of LV functional recovery and (d) the risk of death and HF after STEMI. Importantly, the association with clinical outcomes is the foundation of practicing stratified medicine in patients with STEMI that aims at risk stratification according to the presence of CMD at the end of a successful primary PCI [67]. 

Most of the studies linking CMD with increased risk of mortality and heart failure after STEMI have used thermodilution-based IMR (Table 1). As a consequence, IMR has been used to guide the development of adjunctive therapies following primary PCI, both as a theragnostic maker, i.e., a predictor of therapeutic response, and a measure of therapeutic efficacy [67]. At least five different therapeutic approaches that target post-primary PCI CMD directly, or as a part of a broader concept of reperfusion injury and no-reflow, have been so far investigated (Table 2). First, intracoronary fibrinolytics and different antithrombotic therapies have been tested against the backdrop of distal thrombotic embolization and microvascular plugging as a purported cause of MVO (Figure 1). Initial studies showed a reduction in CFR and IMR at day 2 after primary PCI in patients treated with low-dose intracoronary streptokinase [68]. However, the hitherto largest study (T-TIME) randomized 440 patients with STEMI to intracoronary infusion over 5–10 min of alteplase at a dose of 20 mg vs. 10 mg vs. placebo, immediately after IRA recanalization and before stenting [69]. There was no difference in terms of the primary endpoint of the extent of MVO, expressed as % of LV mass, between either of the two alteplase doses and placebo, nor was there any benefit in terms of clinical outcomes [69]. Moreover, in patients with ischemic times >4 h, intracoronary alteplase was associated with a more extensive MVO as compared with placebo, thus suggesting harm in this patient subgroup [70]. Second, following a similar rationale that recognizes distal embolization as one of the main contributors to the eventual occurrence of MVO, deferred stenting aims at establishing coronary flow in the IRA while deferring stenting to avoid further embolization of the thrombotic material into the vulnerable microvascular bed in the setting of STEMI. Several studies showed that immediate IRA stenting is not associated with improved microcirculatory response in at least a third of patients [71], and that a controlled IRA recanalization with delayed stenting may be associated with more favorable results in terms of the microcirculatory function [72]. However, caution should be warranted with the strategy of deferred stenting, as the large SALVAGE trial (Deferred or Immediate Stent Implantation Based on Microvascular Function in STEMI) was prematurely terminated after enrolling 629 patients due to safety concerns (NCT03581513). Third, vasodilating agents have been used to address microvascular vasoconstriction as another potential cause of CMD in STEMI. When given intracoronary, nicorandil, an ATP-sensitive potassium channels opener and NO donor, was associated with a greater reduction in IMR when compared with nitroglycerine after primary PCI [73]. Recently, the CHANGE trial (Effects of Nicorandil Administration on Infarct Size in Patients With ST-Segment–Elevation Myocardial Infarction Undergoing Primary Percutaneous Coronary Intervention) randomized 238 patients to intravenous nicorandil infusion started prior to primary PCI and continued for 24 h vs. placebo [74]. Intravenous nicorandil was associated with a reduction in CMR-assessed infarct size at 6 months (22.1 ± 11.4% of LV vs. 27.1 ± 13.4%, *p* = 0.005). This finding was strengthened by the observations of less frequent angiographic no-reflow and less MVO on CMR at 5–7 days in patients receiving nicorandil [74]. Although in agreement with some of the previous smaller-scale randomized trials [75], the results favoring nicorandil may need to be interpreted with caution in the context of the previously negative large J-WIND trial [76]. Other vasodilating agents such as adenosine and sodium nitroprusside have not proved beneficial in terms of infarct size and MVO reduction when compared with placebo in a randomized clinical trial including 247 STEMI patients [77]. Fourth, a novel, device-based approach termed pressure-intermittent coronary sinus occlusion (PICSO) has been proposed to affect coronary microvasculature in patients with STEMI. This technique relies on the principle of intermittent coronary sinus occlusion that pushes and redistributes blood backwards including the border zones of the infarcted myocardium. Early applications in humans have confirmed the ability of PICSO to impact the hemodynamics of epicardial arteries by elevating the coronary wedge pressure [78]. Subsequent observational data showed a reduction in infarct size on CMR at day 5 post-anterior STEMI [79]. Importantly, when IMR > 40 was used to risk stratify patients, PICSO was associated with reduced infarct size at 6 months and also a reduction in IMR as compared with a historical control of patients with documented IMR > 40 at the end of primary PCI [80]. The fifth area of research focuses on holistic approaches that are based not only on CMD but on a broader concept of reperfusion injury that aims at cardioprotection at the cardiomyocyte level [3]. Preconditioning (or postconditioning) the myocardium with short episodes of ischemia has shown a potential to reduce infarct size in experimental models [3]. Translating these concepts into the clinical arena, remote ischemic conditioning (inducing intermittent upper limb ischemia via four 5 min blood-pressure cuff inflations) was shown to increase salvage area, expressed as % of LV ventricle, as compared with placebo, albeit no difference in the final infarct size was observed (median 7% of the LV, IQR 1–20, for both groups, *p* = 0.94) [81]. Furthermore, a recent large, randomized, clinical outcome-based trial failed to prove the clinical benefit of remote ischemic conditioning [82]. 

Overall, the described therapeutic approaches targeting CMD in patients with STEMI have been grounded in its multifactorial pathophysiology (Figure 1) and have mainly been tested in terms of their ability to reduce infarct size, without so far being able to translate some of the initial positive effects into a clear clinical benefit (Table 2). 

## 5. Gaps in Knowledge and Future Directions

As the quest for effective therapies continues that may improve the prognosis of STEMI patients beyond the effects of primary PCI and specifically targeting CMD, it is important to understand the constraints of the present evidence base and the debates regarding future directions. First, recent studies in patients with chronic coronary syndrome demonstrated post-PCI values of iFR (instantaneous wave-free ratio) and FFR (fractional flow reserve) to be suboptimal in up to two-thirds and below ischemic threshold in 25–30% of patients despite optimal angiographic results [86,87], thus suggesting that inadequate stenting may be more frequent than what it could be previously derived from angiography only. The so-far discussed mechanisms behind those observations included inadequate stent expansion and residual significant atherosclerotic disease in the culprit vessel. Both may go unnoticed on angiography and have an impact on patient prognosis, independently of CMD, which has not been accounted for in most of the studies described in the present review. 

Second, the invasive assessment of CMD in the catheterization lab (both wire-based and most recently coronary angiography-derived) offers a means of risk stratification immediately at the end of a successful primary PCI. However, the actionability of these measurements may be complicated by (a) the heterogeneity of the evidence base and the practical aspects of obtaining invasive indices of CMD, (b) the dynamic nature of microvascular damage in STEMI and (c) the multifactorial cause of CMD rendering therapeutic targeting difficult. 

### 5.1. Heterogeneity and Practicality of The Invasive Assessment of CMD 

Correlation coefficients linking IMR, the most frequently studied invasive CMD index, with the CMR-based MVO, have ranged from 0.2 to 0.8 [88], with pooled data analyses showing the highest likelihood of MVO with the IMR cut-off values >41 [89] and >46 [88]. Furthermore, the recent comparative analysis suggested that PzF and HMR may be superior to IMR in predicting MVO and infarct size [62]. However, high-quality simultaneous pressure and flow tracings may be difficult and time consuming to obtain with the Doppler wire, and PzF calculation requires offline post-processing with dedicated software, thus limiting the practical applicability of HMR and PzF. On the other hand, manual saline injections introduce a degree of variability in the measurement of IMR. Moreover, and as discussed earlier in Section 3.2, several early investigations confirmed the dependency of both IMR and HMR measurements on the presence of epicardial stenosis, which can be accounted for if collateral flow is factored in the resistance calculation [34,90]. However, most of the studies in patients with STEMI have assumed that primary PCI would render the culprit artery without significant epicardial flow impedance and have thus not reported data on the need to correct their IMR/HMR measurements for collateral flow. 

Overall, despite the described shortcomings, the consistent association with clinical events, mainly cardiac death and heart failure, demonstrated for the IMR cut-off > 40 [88], seems to, at present, offer the possibility to progress with developments of therapeutic strategies that would deliberately target patients with IMR > 40 at the end of primary PCI [67].

### 5.2. Time-Course of Microvascular Damage in STEMI

Another challenge of treating CMD in STEMI is the dynamic nature of microvascular damage and reperfusion injury. Traditionally, invasive assessment of CMD has been performed immediately after primary PCI, albeit several studies reported repeated measurements. Early studies that followed the time course of CFR showed that it increases from mean values of 1.3–1.5 immediately after IRA recanalization to values > 2 after approximately two weeks and up to 6 months after STEMI [91,92]. Importantly, the extent of MVO on CMR did not change between days 2 and 9 [93], and the increase in CFR was unrelated to myocardial contractility or the perfusion defect on imaging [91,92]. Recent serial CMR assessments showed progression of myocardial hemorrhage and edema (in those without hemorrhage) over the first 10 days post-MI with restitution in most cases by 7 months [94]. In patients with myocardial hemorrhage (43% of the enrolled population at peak day 3), adverse LV remodeling was noted in the follow-up as evidenced by an increase in LV end-diastolic volume [94]. A more recent study linked the extent of CFR and IMR recovery over the 24 h after primary PCI with myocardial salvage index and LV ejection fraction at 6 months [95]. Interestingly, the observed dynamics in CFR and IMR values over the first 24 h appear to have been more pronounced in patients with evidence of MVO on CMR [96]. Another study linked the improvement in CFR and IMR from day 2 until 5 months with a significant reduction in infarct size [97]. Similar dynamics were also reported for HMR, which decreased significantly over the first 7 days after primary PCI [98]. The described evolution in the function of coronary microvasculature in patients with STEMI, although more pronounced in the infarct territory, also pertains to non-IRA territories where temporal studies have shown improvements in CFR and HMR over the first month [99]. 

Of note, the overall imaging evidence suggests that MVO remains constant in the first 3 days after STEMI, with gradual restitution in the following days and weeks and persistent MVO in 10–15% of patients at 6 months [100]. These dynamics need to be aligned with the invasive CMD assessment and taken into account when planning the timing of adjunctive therapies. 

### 5.3. Multifactorial Nature of Microvascular Damage in STEMI

One of the main constraints in developing successful treatment strategies targeting CMD in STEMI appears to reside in its multifactorial pathophysiology (Figure 1). Most of the therapies that progressed into the clinical arena (Table 2) targeted individual pathophysiological axes and/or were based on the experimental models showing infarct size reduction [100]. The latter point is important as MVO has been associated with poor prognosis after STEMI above and beyond infarct size. Moreover, targeting patients with proof of increased microvascular resistance, as opposed to all comers, may improve the effect size [67]. An example of this principle is the concept that targeting residual myocardial iron, which has been associated with adverse LV remodeling post-MI, may improve outcomes. Iron chelation with deferoxamine was thus tested in a small randomized trial which failed to show a reduction in infarct size despite the achieved reduction in oxidative stress [101]. However, the effect of iron chelation may have been diluted in an all-comer population, and stratified inclusion based on the occurrence of myocardial hemorrhage may have yielded different results. 

## 6. Conclusions

Microvascular injury is detected in half of patients with angiographically successful primary PCI, and it is associated with poor prognosis after STEMI above and beyond infarct size. Invasive assessment of CMD in the catheterization laboratory, wire-based or recently developing coronary angiography-derived, has been shown to correlate with MVO on CMR and to predict cardiac death and heart failure, making it thus a valuable risk stratification tool. Given the relative lack of success with the hitherto tested treatment strategies that were aiming at restoring microvascular function, early risk stratification based on invasive indices of CMD in the catheterization laboratory may provide new ground for the ongoing research efforts to improve the long-term prognosis of patients with STEMI. 

## Figures and Tables

**Figure 1 jcm-12-01602-f001:**
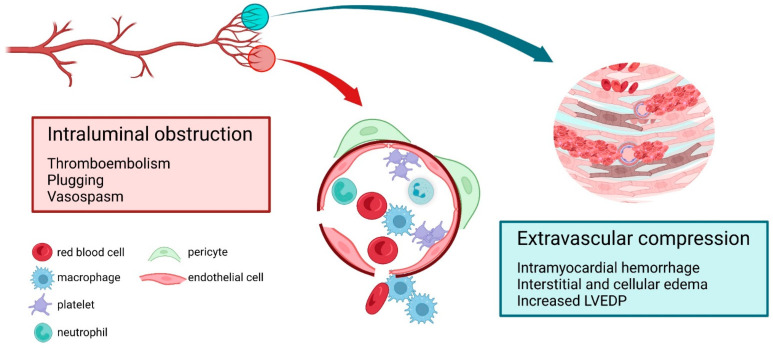
Pathophysiological mechanisms contributing to coronary microvascular dysfunction in patients with acute myocardial infarction. Created with BioRender.

**Figure 2 jcm-12-01602-f002:**
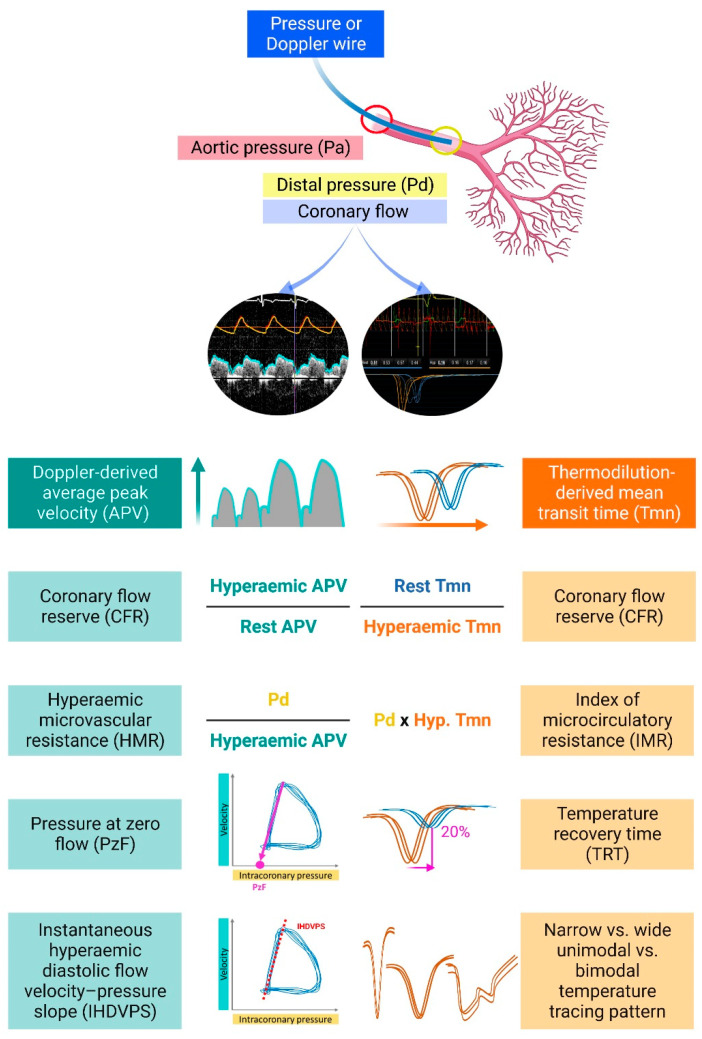
Different approaches to invasive assessment of coronary microvascular dysfunction in the catheterization laboratory. Created with BioRender.

**Table 1 jcm-12-01602-t001:** Selected studies showing association of different invasive parameters of microvascular resistance with clinical outcomes in patients with acute myocardial infarction treated with primary PCI.

Study	Year	Sample Size	Index Cutoff	Primary (Composite) Clinical Endpoint	Event Rate vs. No CMD Group	Adjusted HR (95% CI)	Follow-Up Duration
Takahashi et al. [31]	2007	118	CFR ≤ 1.3	Cardiac death, HF or recurrent MI	40% vs. 4%	-	62 months
Fearon et al. [51]	2013	253	IMR > 40	Death or HF hospitalization	20.0% vs. 11.0%	2.2 (1.1–4.5)	33 months
Van de Hoef [32]	2013	100	CFR < 1.5	Cardiac mortality	20.0% vs. 9.0%	1.6 (0.5–5.0)	120 months
Fukunaga et al. [37]	2014	88	Bimodal pattern (thermodilution)	Cardiac death, nonfatal MI and HF rehospitalization	73.3% vs. 7.3%	27.8 (2.4–320.5)	6 months
Jin et al. [53]	2015	145	HMR > 2.8	Cardiac death or HF hospitalization	17.2% overall	1.7 (1.3–2.3)	85 months
Carrick et al. [15]	2016	283	IMR > 40	All-cause death or first HF event after index hospitalization	3.5% overall	4.4 (1.9–10.1) ^†^	28 months
Fahrni et al. [50]	2017	261	IMR > 40	Major cardiac complications ^•^	16.7% vs. 0%	-	1 month
De Waard et al. [54]	2018	176	HMR ≥ 3.0 CFR < 1.5	All-cause death or HF hospitalization	20.3% vs. 3.6% 14.9% vs. 4.5%	7.0 (1.5–33.7) 3.5 (1.1–10.8)	38 months
Yew et al. [49]	2018	278	Bimodal pattern (thermodilution)	All-cause death or HF hospitalization	14.4% overall	2.3 (0.9–6.1)	48 months
Maznyczka et al. [40]	2020	144	IMR > 140 RRR ≤ 1.7	Cardiac death, nonfatal MI, HF hospiltalization	28.1% vs. 8.0% 20.8% vs. 10.4%	4.4 (1.7–11.7) ^◊^ 2.2 (0.8–5.8) ^◊^	12 months
Scarsini et al. [55]	2021	198	IMR > 40	All-cause mortality, HF, resuscitated cardiac arrest, malignant ventricular arrhythmias or the need for a primary prevention ICD	25.5% * vs. 5.7% 22.2% ° vs. 5.7%	4.6 (1.4–16.1) * 6.8 (1.8–25.2) °	40 months
Maznyczka et al. [48]	2021	271 ^‡^ 144 ^‖^	TRT > 0.5	All-cause death or HF hospitalization	19.2% overall 15.0% overall	5.4 (2.0–14.4) 5.8 (1.4–23.9)	60 months 12 months
Yoon et al. [52]	2021	326	IMR > 29	All-cause death or HF hospitalization	10.3% vs. 2.1%	4.0 (1.2–12.9)	65 months
Choi et al. [56]	2021	309	Angio-IMR > 40	Cardiac death or HF hospitalization	46.7% vs. 16.6%	2.2 (1.2–4.1)	120 months

* IMR > 40 with concomitant MVO on CMR. ° IMR > 40 without concomitant MVO on CMR. ^◊^ From the univariate regression analysis. ^†^ Based on the model with the endpoint including death and HF during index hospitalization. ^‖^ Derivation cohort for TRT. ^‡^ Validation cohort for TRT. ^•^ cardiac death, cardiogenic shock, documented pulmonary edema, malignant ventricular tachycardia, malignant bradyarrhythmia, cardiac wall rupture, and intraventricular thrombus. PCI—percutaneous coronary intervention. HF—heart failure. ICD—implantable cardioverter defibrillator. CMD—coronary microvascular dysfunction. HR—hazard ratio. CI—confidence interval. IMR—index of microcirculatory resistance. HMR—hyperemic microvascular resistance. RRR—resistive reserve ratio. TRT—temperature recovery time.

**Table 2 jcm-12-01602-t002:** Overview of different therapeutic strategies targeting coronary microvascular dysfunction after STEMI with representative trials depicting translation from microcirculation- and infarct size- to-clinical outcomes-based studies.

Therapeutic Strategy	Study	Year	Design	Sample Size	Comparison	Primary Endpoint Intervention vs. Control	Overall Outcome
Intracoronary fibrinolytics	Sezer et al. [68]	2007	RCT	41	IC streptokinase (250 kU) vs. placebo	CFR: 2.0 vs. 1.4, *p* = 0.002 IMR: 16.3 vs. 32.5, *p* < 0.01	Improvement in IMR and CFR, but no effect on infarct size
	McCartney et al. [69]	2019	RCT	440	IC alteplase (20 mg) * vs. placebo	MVO on CMR as %LV 3.5% vs. 2.3%, *p* = NS	No reduction in MVO, even harmful in patients presenting >4 h of symptom onset [70]
Antiplatelets	Kirma et al. [83]	2012	RCT	49	IC vs. IV tirofiban	IMR/CFR at day 5 IMR: 27 vs. 35, *p* = NS CFR: 2.2 vs. 1.9, *p* = NS	No difference in the effect on microcirculation
	Stone et al. [84]	2012	RCT	353 °	IC abciximab vs. placebo	Infarct size 15.1% vs. 17.9%, *p* = 0.03	Reduction in infarct size with IC abciximab
	Thiele et al. [85]	2012	RCT	2065	IC vs. IV bolus + infusion abciximab	Death, MI or HF 7.0 vs. 7.6%, *p* = NS	No difference in clinical outcomes between IV and IC abciximab
Deferred stenting	Sezer et al. [72]	2022	RCT	20	Delayed vs. immediate stenting	Pressure at zero flow 41.5 vs. 76.9, *p* = 0.001	Delaying stenting for 30 min after IRA recanalization was associated with less microvascular injury
	NCT03581513	2022	RCT	880	Deferred vs. immediate stenting	All-cause death, HF, MI and target vessel revascularization	Prematurely terminated due to safety concerns upon inclusion of 629 patients
Vasodilating agents	Ito et al. [73]	2013	RCT	60	IC nicorandil vs. nitroglycerine	Reduction in IMR 10 vs. 2, *p* = 0.0002	Nicorandil IC reduces CMD assessed by IMR
	Qian et al. [74]	2022	RCT	238	IV nicorandil vs. placebo	Infarct size on CMR 19.5 g vs. 25.7 g, *p* = 0.008	Nicorandil IV given prior to reperfusion reduced infarct size
	Kitakaze et al. [76]	2007	RCT	1216	IV ANP vs. placebo IV nicorandil vs. placebo	Infarct size (CK IU/mL) 66.45 vs. 77.78, *p* = 0.016 70.52 vs. 70.85, *p* = NS	IV ANP but not nicorandil decreased CK release and improved LVEF in the follow-up
	Nazir et al. [77]	2016	RCT	247	IC adenosine vs. SNP vs. control	Infarct size on CMR 10% vs. 10% vs. 8%, *p* = NS	Neither adenosine nor SNP reduced infarct size or MVO
Device therapy	De Maria et al. [80]	2018	Obs.	105	PICSO vs. control	IMR/infarct size on CMR IMR: 24.8 vs. 45.0, *p* < 0.01 IS: 26% vs. 33%, *p* = 0.006	PICSO reduced IMR and IS in patients with IMR > 40 at the end of primary PCI
	Botker et al. [81]	2010	RCT	333	RIC vs. control	Myocardial salvage index 0.75 vs. 0.55, *p* = 0.03	Remote conditioning by intermittent arm ischemia increased the proportion of salvaged myocardial area at risk
	Hausenloy [82]	2019	RCT	5401	RIC vs. control	Cardiac death or HF 9.4% vs. 8.6%	Remote conditioning by intermittent arm ischemia did not impact clinical outcomes

* The trial randomized patients to either intracoronary alteplase 20 mg or 10 mg or placebo, but the comparison between the 20 mg dose and placebo was considered primary. ° The INFUSE-AMI trial randomized 452 patients in a 2 × 2 factorial design to IC abciximab vs. no abciximab and manual thrombectomy vs. no thrombectomy. 352 patients were ultimately evaluated for the effect of abciximab. STEMI—ST-segment elevation myocardial infarction. RCT—randomized controlled trial. HF—heart failure. Obs.—observational study. IC—intracoronary. IV—intravenous. MVO—microvascular obstruction. CMR—cardiac magnetic resonance. IMR—index of microcirculatory resistance. CMD—coronary microvascular dysfunction. ANP—atrial natriuretic peptide. CK—creatine kinase. LVEF—left ventricular ejection fraction. SNP—sodium nitroprusside. PICSO—pressure-controlled intermittent coronary sinus occlusion. IS—infarct size. RIC—remote ischemic conditioning.

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
