# Peer review of "Coronary Microcirculation: The Next Frontier in the Management of STEMI"

_jcm, 2023, doi:10.3390/jcm12041602_

Round 1

Reviewer 1 Report

Dear authors,

The review “Coronary microcirculation: the next frontier in the management of STEMI” is an important detailed review on microvascular injury after STEMI with angiography successful primary PCI. The work described the contemporary collected data on microvascular disfunction following STEMI, outcomes after primary PCI, the practical role of invasive assessment of coronary microvascular disfunction and therapeutic strategies targeting this disfunction.

I believe it will be helpful for interventional and non-interventional cardiologists.  However, the main question needs to be addressed:

Questions/Comments:

Major issues:

1.  1.   Do the authors believe that “angiography successful primary PCI” excludes the macrovascular disfunction in a target-artery? The authors described that there were about 50% patients with impaired invasive coronary parameters after STEMI and appropriate PCI. It was the proofed evidence, that these impaired parameters were associated with poor prognosis. However, the most part of the referenced studies, which had shown such data, performed on patient populations in 2008-2013 years (reference 11), 2011-2012 years (reference 15, 47, 49), 2005-2015 years (reference 50, 51) etc.  These cohorts were stented with angiography guided percutaneous coronary intervention. The hitherto collected evidence demonstrates that over two-thirds of patients had a physiologically suboptimal or poor result after angiography-guided PCI [1]. The great pool of patients with “optimal PCI results” by conventional angiography has an inadequately deployed stents as well as high atherosclerotic burden elsewhere in the artery and it is associated with poor prognosis [2]. So, these cohorts, likely, contained patients both with microvascular and with macrovascular disfunctions. I guess, this major limitation of the previous studies should be particularly addressed in the review.

22.   IMR and HMR may not reflect the isolated contribution of microvascular resistance to coronary flow impedance. Some invasive studies substituting coronary flow velocity for myocardial blood flow have suggested that microvascular resistance paradoxically increases with increasing stenosis severity [3-5]. The IMR’s ability to estimate microvascular resistance independently of the degree of epicardial stenosis was demonstrated only, when corrected for the collateral flow. However, some studies in references didn’t point how they measured IMR in details. So, this parameter could be partly or mostly affected by epicardial coronary flow. 

References:

11.Collison D, Didagelos M, Aetesam-Ur-Rahman M, Copt S, McDade R, McCartney P, Ford TJ, McClure J, Lindsay M, Shaukat A, Rocchiccioli P, Brogan R, Watkins S, McEntegart M, Good R, Robertson K, O'Boyle P, Davie A, Khan A, Hood S, Eteiba H, Berry C, Oldroyd KG. Post-stenting fractional flow reserve vs coronary angiography for optimization of percutaneous coronary intervention (TARGET-FFR). Eur Heart J. 2021 Dec 1;42(45):4656-4668. doi: 10.1093/eurheartj/ehab449.

22. Patel M, Jeremias A, Maehara A, et al. 1-Year Outcomes of Blinded Physiological Assessment of Residual Ischemia After Successful PCI. J Am Coll Cardiol Intv. 2022 Jan, 15 (1) 52–61. https://doi.org/10.1016/j.jcin.2021.09.042

33. Sambuceti G, Marzilli M, Fedele S, et al. Paradoxical increase in microvascular resistance during tachycardia downstream from a severe stenosis in patients with coronary artery disease: reversal by angioplasty. Circulation. 2001; 103: 2352–2360.

4 4. Chamuleau SAJ, Siebes M, Meuwissen M, et al. The association between coronary lesion severity and distal microvascular resistance in patients with coronary artery disease. Am J Physiol Heart Circ Physiol. 2003; 285: H2194–H2200.

  5. Marzilli M, Sambuceti G, Fedele S, et al. Coronary microcirculatory vasoconstriction during ischemia in patients with unstable angina. J Am Coll Cardiol. 2000; 35: 327–334

Author Response

Dear reviewer, thank you very much for your in-depth analysis of our paper and the important comments made, which help us improve our manuscript. We have added the relevant information to the manuscript as well as inserted the suggested references. Please find below point-by-point answers to your comments. Our answers are in bold, and we quote the changes in the manuscript based on your comments in italic. 

1. Do the authors believe that “angiography successful primary PCI” excludes the macrovascular disfunction in a target-artery? The authors described that there were about 50% patients with impaired invasive coronary parameters after STEMI and appropriate PCI. It was the proofed evidence, that these impaired parameters were associated with poor prognosis. However, the most part of the referenced studies, which had shown such data, performed on patient populations in 2008-2013 years (reference 11), 2011-2012 years (reference 15, 47, 49), 2005-2015 years (reference 50, 51) etc.  These cohorts were stented with angiography guided percutaneous coronary intervention. The hitherto collected evidence demonstrates that over two-thirds of patients had a physiologically suboptimal or poor result after angiography-guided PCI [1]. The great pool of patients with “optimal PCI results” by conventional angiography has an inadequately deployed stents as well as high atherosclerotic burden elsewhere in the artery and it is associated with poor prognosis [2]. So, these cohorts, likely, contained patients both with microvascular and with macrovascular disfunctions. I guess, this major limitation of the previous studies should be particularly addressed in the review. 

We agree with this comment emphasizing recent data on inadequate physiological result after what appears to be angiographically optical stenting result. Acknowledging that most of the so far collected data (TARGET FFR and DEFINE PCI studies) are in patients presenting with chronic coronary syndromes, we agree that not achieving otpimal stenting results may also impact prognosis of STEMI patients above and beyond microcirculatory function. Therefore, we add the following paragraph to the manuscript on pages 16 and 17, highlighted in yellow. 

First, recent studies in patients with chronic coronary syndrome demonstrated post-PCI values of iFR (instantaneous wave-free ratio) and FFR (fractional flow reserve) to be sub-optimal in up to two thirds and below ischemic threshold in 25-30% of patients despite optimal angiographic results [81, 82], thus suggesting that inadequate stenting may be more frequent than what it could be previously derived from angiography only. The so far discussed mechanisms behind those observations included inadequate stent expansion and residual significant atherosclerotic disease in the culprit vessel. Both may go unnoticed on angiography and have an impact on patient prognosis, independently of CMD, which has not be accounted for in most of the studies described in the present review."

2. IMR and HMR may not reflect the isolated contribution of microvascular resistance to coronary flow impedance. Some invasive studies substituting coronary flow velocity for myocardial blood flow have suggested that microvascular resistance paradoxically increases with increasing stenosis severity [3-5]. The IMR’s ability to estimate microvascular resistance independently of the degree of epicardial stenosis was demonstrated only, when corrected for the collateral flow. However, some studies in references didn’t point how they measured IMR in details. So, this parameter could be partly or mostly affected by epicardial coronary flow. 

Thank you very much for this important comment with which we agree. Consequently we have added the following paragraph on pages 17 and 18, highlighted in yellow. 

"Moreover, and as discussed earlier in section 3.2., several early investigations confirmed dependency of both IMR and HMR measurements on the presence of epicardial stenosis, which can be accounted for if collateral flow is factored in the resistance calculation [33, 85]. However, most of the studies in patients with STEMI have assumed that primary PCI would render the culprit artery without significant epicardial flow impedance and have thus not reported data on the need to correct their IMR/HMR measurements for collateral flow."

Reviewer 2 Report

This is a comprehensive review on the available invasive means of assessment of the coronary microvascular obstruction following primary PCI. 

However, a drawback is represented by the omission from this review of an invasive parameter, coronary wedge pressure (CWP) that correlates with MVO and myocardial remodelling after STEMI (please refer to the doi:10.1038/s41598-018-20276-6Pre-revascularization coronary wedge pressure as marker of adverse long-term left ventricular remodelling in patients with acute ST-segment elevation myocardial infarction).

Otherwise, it is a well written article, including clear and evocative figures, and recent references.

Author Response

Dear reviewer, thank you very much for your careful analysis of our manuscript. We agree with your comments and we have inserted the following paragraph on page 6, highlighted in yellow. We have also inserted the suggested reference to the manuscript. Thank you. 

"Going beyond angiographic markers such as TIMI flow and MBG, invasive assessments of coronary microcirculation have been used to assess microcirculatory function. Recently, increased coronary wedge pressure (CWP>38 mmHg), as measured through a microcatheter advanced distal to the occlusion site and before culprit artery recanalization, was associated with adverse LV remodelling over 60 months follow-up after STEMI [23]."

Round 2

Reviewer 1 Report

Authors have adequately addressed all the issues.

Reviewer 2 Report

The manuscript was revised according to my comments.